# The Clinical Frailty Scale: Estimating the Prevalence of Frailty in Older Patients Hospitalised with COVID-19. The COPE Study

**DOI:** 10.3390/geriatrics5030058

**Published:** 2020-09-21

**Authors:** Jemima T. Collins, Roxanna Short, Ben Carter, Alessia Verduri, Phyo K. Myint, Terence J. Quinn, Arturo Vilches-Moraga, Michael J. Stechman, Susan Moug, Kathryn McCarthy, Jonathan Hewitt

**Affiliations:** 1Department of Geriatric Medicine, Aneurin Bevan UHB, Caerphilly CF82 7GP, UK; jemimacollins@doctors.net.uk; 2Department of Forensic and Neurodevelopmental Sciences, King’s College London, London SE5 8AF, UK; Roxanna.short@kcl.ac.uk; 3Department of Biostatistics and Health Informatics, King’s College London, London SE5 8AF, UK; ben.carter@kcl.ac.uk; 4Respiratory Unit, Hospital Policlinico, University of Modena and Reggio Emilia, 41121 Modena, Italy; VerduriA@cardiff.ac.uk; 5Institute of Applied Health Sciences, University of Aberdeen, Aberdeen AB25 2ZD, UK; phyo.myint@abdn.ac.uk; 6Institute of Cardiovascular and Medical Sciences, University of Glasgow, Glasgow G12 8TA, UK; terry.quinn@glasgow.ac.uk; 7Ageing and Complex Medicine Department, Salford Royal NHS Trust, University of Manchester, Manchester M6 8HD, UK; Arturo.vilches-moraga@srft.nhs.uk; 8Department of Surgery, Cardiff and Vale UHB, Cardiff CF14 4XW, UK; Michael.stechman@wales.nhs.uk; 9Department of Surgery, Royal Alexandra Hospital, Paisley PA2 9PN, UK; susanmoug@nhs.net; 10Department of Surgery, Southmead Hospital, North Bristol NHS Trust, Bristol BS10 5NB, UK; Kathryn.mccarthy@nbt.nhs.uk; 11Division of Population Medicine, Aneurin Bevan UHB, Cardiff University, Cardiff CF14 4XN, UK

**Keywords:** frailty, Clinical Frailty Scale, COVID-19, prevalence, hospital

## Abstract

Frailty assessed using Clinical Frailty Scale (CFS) is a good predictor of adverse clinical events including mortality in older people. CFS is also an essential criterion for determining ceilings of care in people with COVID-19. Our aims were to assess the prevalence of frailty in older patients hospitalised with COVID-19, their sex and age distribution, and the completion rate of the CFS tool in evaluating frailty. **Methods:** Data were collected from thirteen sites. CFS was assessed routinely at the time of admission to hospital and ranged from 1 (very fit) to 9 (terminally ill). The completion rate of the CFS was assessed. The presence of major comorbidities such as diabetes and cardiovascular disease was noted. **Results:** A total of 1277 older patients with COVID-19, aged ≥ 65 (79.9 ± 8.1) years were included in the study, with 98.5% having fully completed CFS. The total prevalence of frailty (CFS ≥ 5) was 66.9%, being higher in women than men (75.2% vs. 59.4%, *p* < 0.001). Frailty was found in 161 (44%) patients aged 65–74 years, 352 (69%) in 75–84 years, and 341 (85%) in ≥85 years groups, and increased across the age groups (<0.0001, test for trend). Conclusion: Frailty was prevalent in our cohort of older people admitted to hospital with COVID-19. This indicates that older people who are also frail, who go on to contract COVID-19 may have disease severity significant enough to warrant hospitalization. These data may help inform health care planners and targeted interventions and appropriate management for the frail older person.

## 1. Introduction

Frailty is a pivotal factor in determining risk of poor health-related outcomes. It confers an increased vulnerability to non-restoration of homeostasis after a stressor event [1]. Simply put, it means that even a minor insult may result in hospitalisation and death. The Clinical Frailty Scale (CFS) is widely used to assess frailty [2], is easy to use, and well-validated in non-COVID-19 populations aged 65 years and over [3].

COVID-19 has affected older people proportionately more than younger people [4]. At the start of this disease, public health guidance was for all people aged 70 and older to shield or self-isolate. As the pandemic continues, new approaches to assessing risk may be required, distinct from a blanket policy to shield all people over a certain age. This may include the assessment of frailty, in order to better allocate care resources to people at greater risk of negative outcomes and identify potential targets for preventative interventions. For example, muscle strength training and protein supplementation may delay or even reverse the progression of frailty in older people [5,6].

Current estimates of frailty are still evolving as learning about COVID-19 continues. Any estimate is also likely to vary according to the method of frailty assessment and the population which is studied. In this European multi-centre cohort study—the COVID-19 in Older People (COPE) study —we aim to describe frailty prevalence estimates in more detail in older people hospitalised with COVID-19, their sex and age distribution, and the completion rate of the CFS.

## 2. Materials and Methods

### 2.1. Study Design

The COPE study primary aims are to evaluate the association of frailty with clinical outcomes and mortality in patients admitted to hospital with COVID-19; a full study protocol can be found elsewhere [7]. Following appropriate ethical permissions, two rounds of data collection were performed retrospectively between 27th February and 9th June 2020. CFS was collected routinely as per NICE guidelines on admission to hospital [8].

### 2.2. Setting

Data collection was performed in twelve UK sites and one Italian site, via an established network of clinicians with an interest in frailty (www.opsoc.eu).

### 2.3. Participants

We included all unselected sequential patients admitted to hospital with COVID-19. Frailty was assessed using the CFS on admission by the admitting clinical team or by the research investigator overseen by the Principal Investigator at each site. Criteria for diagnosis were SARS-CoV-2 positive swabs, or a clinical diagnosis consistent with COVID-19. Site parent or research clinical teams screened in-patient admission lists for eligibility and observed local data protection policies.

### 2.4. Variables

Demographic and biochemical categorical variables expressed in numbers (percentage) with prognostic utility [9,10,11] were gathered and analysed. This included: age, sex, ethnicity, C-reactive protein (CRP), estimated glomerular filtration rate (eGFR) calculated using the Modification of Diet in Renal Disease (MDRD) equation, and albumin. Comorbidity factors included smoking status, coronary artery disease, diabetes mellitus, hypertension, chronic obstructive pulmonary disease (COPD), and cardiac failure. Age groups were categorised into 65–74 years, 75–84 years, and 85 years and over.

The CFS is an appraisal of functional and cognitive status two weeks prior to hospitalization. This numerical scale corresponds with increasing severity, with 1 being very fit, 2 well, 3 managing well, 4 vulnerable, 5 mildly frail, 6 moderately frail, 7 severely frail, 8 very severely frail and 9 terminally ill. The completion rate was calculated for the CFS.

### 2.5. Data Analysis

The prevalence of frailty by CFS categories was presented descriptively in tabular format, according to demographics (Table 1) and age group (Table 2). A test for trend for increasing frailty was conducted for increasing frailty within each age group. CFS of 5 and above was considered to be frail [12,13]. Analysis was carried out using Stata v15 [14].

## 3. Results

A total of 1277 patients were eligible and all were aged 65 years and above with a positive diagnosis of COVID-19. The sex distribution was 52.9% male and 47.1% female. CFS was fully completed in 98.5% of all patients. The mean (SD) age was 79.9 (8.1) years. The total prevalence of frailty (CFS ≥ 5) was 66.9%. There was heterogeneity between the CFS across the sites, but patients were overwhelmingly of white ethnicity (Table 1).

The number (%) of patients with CFS = 1 was 15 (1.2%) CFS = 2 63 (5.0%), CFS = 3 169 (13.5%), CFS = 4 157 (12.6%), CFS = 5 186 (14.9%), CFS = 6 272 (21.8%), CFS = 7 289 (23.1%), CFS = 8 80 (6.4%) and CFS = 9 46 (3.6%).

Within our population, the prevalence of frailty was 75.2% in women, and 59.4% in men (*p* < 0.001). The prevalence of frailty within these three age groups was 44.6%, 69.7% and 86.9%. Frailty increased with advancing age (*p* < 0.0001, test for trend). The CFS was fully completed in 98.5% of participants (Table 2).

The prevalence of major comorbidities in our cohort was as follows: diabetes (28.0%), coronary artery disease (26.9%), hypertension (56.1%), COPD (14.5%), and heart failure (12.6%).

## 4. Discussion

Our results show that the prevalence of frailty is high (66.9%), when assessed using the CFS in a population aged 65 years and above hospitalised with COVID-19. Frailty increased with increasing age and was greater in females compared to males (both *p* < 0.001). Both these trends are similar to previous reports in community-dwelling older people [15], but the overall prevalence is much lower at 9.9–43.7% [15,16]. This underlines how crucial assessment of frailty is, when considering the risk of COVID-19 and subsequent management, and is likely to become more important, in the event of a further spike in cases.

Frailty is not yet well described in COVID-19. The CFS is an important starting point on which to base clinical judgment in a resource-scarce environment. Clinicians assessing the CFS should understand the concept of frailty and its principles of disability, comorbidity, and cognitive impairment [17], and it is suggested that inter-rater agreement is good [18]. This rapid assessment in UK hospitals may have played a part in appropriate management of individual patients, as per NICE guidelines [8]. The CFS can predict mortality for acute hospitalised older patients [19,20,21]. While the CFS is used in younger populations [22] it is far more widely used and validated in patients aged 65 and over [3], as in our current study, where it demonstrated a very high rate of completion, which would echo these advantages.

The prevalence of frailty (CFS ≥ 5) in our study was high (66.9%), which was similar to recently published results from a cohort comparing non-COVID-19 older patients with COVID-19 patients [23]. These figures are high but unsurprising, given the unselected nature of our cohort who are hospitalised with COVID-19. In-patient estimates of frailty using the CFS in older people have also been high. Hartley and colleagues found that 75% of patients aged 75 and over admitted to medical wards had CFS ≥ 5 [24], and in other hospital-based studies, the prevalence was 36–45% [25,26]. These values contrast with the prevalence of frailty in community-dwelling older people, where previous estimates have ranged from 9.9–13.6% (pooled estimates) [15], to 22.4% and 43.7% over the age of 65, and over the age of 85, respectively [16]. In a non-COVID-19 population, all-cause mortality in the severely frail was 33%, but was much higher in the COVID-19 group at 60% [23]. This implies that frailty is not the sole cause of mortality, rather a key factor for consideration, along with the acute illness [27].

Our study found that more women than men were frail (77.5% vs. 60.1%) (Table 1). There is emerging evidence regarding the protective effect of the X chromosome in females [28], speculatively due to a more robust immune system in females, although the biology of sex differences in COVID-19 is still unclear. Previously, we have shown that increasing frailty by CFS can predict mortality [29]. Although men are more commonly affected by, and hospitalised with COVID-19, women may be frailer than men but survive for longer. Future associations should focus on the interplay between age, sex, and frailty.

The importance of screening older people for frailty, with the intention of identifying those most at risk is fully supported by this work. This would allow clinicians to be better prepared to assess and triage older people in the event of the next pandemic or any other acute illness that may require hospitalization. Due consideration should also be given to older patients who are not yet frail but vulnerable, as to whether they warrant particular public health advice, in both hospital and community settings. Shielding for the duration of this pandemic has been widely advised for all people over the age of 70, but should this be the case for fit and well older people? Chronological age on its own is not a biomarker of poor prognosis, whereas frailty is [30]. Identification of those who are vulnerable (CFS = 4) and have mild frailty (CFS = 5) may be the starting point for clinical trials of muscle strength training and protein supplementation, which have been found to have modest benefit in slowing the progression of frailty in the non-COVID-19 population [5]. At this time, there is still no vaccine or definitive treatment for COVID-19. Therefore, a parallel research focus for older people might be to maximise risk reduction by targeted frailty prevention, and the first step in that process is identification of those most at risk, irrespective of age.

We note that there may have been selection bias as less frail people may have remained out of hospital due to a reduced susceptibility to the condition. Another limitation is although assessment of the CFS was performed and overseen by clinicians with experience at collecting frailty data, we could not account for inter-rater reliability as this was not examined. Furthermore, the impact of social isolating could have resulted in a change of CFS, prior to admission, as older people were likely to be less socially active than they were previously. Finally, our results do not tell the whole story of frailty prevalence in COVID-19, as many frail people are not admitted to hospital, and are palliated in the community. Thus, our results do not account for frailty prevalence in the community setting.

## 5. Conclusions

The completion rate of the CFS was high, in a cohort of older patients admitted to hospital with COVID-19. Frailty prevalence was high in this group, and increasing frailty was associated with increasing age and was more prevalent in females. Every patient over the age of 65 at risk of COVID-19 should be screened for frailty. This may enable measures to be taken whilst in the community, in order to identify targets for slowing progression of frailty and enable optimisation of physical reserves.

## Figures and Tables

**Table 1 geriatrics-05-00058-t001:** Clinical Frailty Scale, demographics, and comorbidities in patients aged 65 or over admitted to 13 sites in UK and Italy—the COVID-19 in Older People (COPE) study.

	CFS 1–4 N = 423	CFS 5–9 N = 854	Total N = 1277
**Sex**			
Female	149 (24.8)	452 (75.2)	601 (47.1)
Male	274 (40.6)	401 (59.4)	675 (52.9)
missing	0	1	1
**Age**			
65–74	205 (56.0)	161 (44.0)	366 (28.7)
75–84	158 (31.0)	352 (69.0)	510 (39.9)
85+	60 (15.0)	341 (85.0)	401 (31.4)
**Ethnicity**			
White	352 (32.3)	738 (67.7)	1090 (85.4)
Asian	7 (35.0)	13 (65.0)	20 (1.6)
Black	2 (40.0)	3 (60.0)	5 (0.4)
Other	2 (25.0)	6 (75.0)	8 (0.6)
Missing	60	94	154
**Smoking**			
Never	203 (32.7)	417 (67.3)	620 (48.6)
Ex-smoker	187 (34.2)	360 (65.8)	547 (42.8)
Current smoker	20 (27.0)	54 (73.0)	74 (5.8)
Missing	13	23	36
**Diabetes**			
Yes	110 (30.8)	247 (69.2)	357 (28.0)
No	312 (34.1)	604 (65.9)	916 (71.7)
Missing	1	3	4
**Coronary Artery Disease**			
Yes	97 (28.2)	247 (71.8)	344 (26.9)
No	324 (34.8)	606 (65.2)	930 (72.8)
Missing	2	1	3
**Hypertension**			
No	174 (31.1)	386 (68.9)	560 (43.9)
Yes	77 (34.2)	148 (65.8)	225 (17.6)
Yes (on treatment)	170 (34.7)	320 (65.3)	490 (38.4)
missing	2	0	2
**CRP**			
>40	315 (35.5)	573 (64.5)	888 (69.5)
<=40	108 (27.8)	281 (72.2)	389 (30.5)
**eGFR**			
>60	257 (38.6)	409 (61.4)	666 (52.2)
45–59	55 (29.3)	133 (70.7)	188 (14.7)
30–44	50 (22.9)	168 (77.1)	218 (17.1)
<30	33 (22.4)	114 (77.6)	147 (11.5)
missing	28	30	58
**Albumin**			
<35	237 (28.9)	583 (71.1)	820 (64.2)
>=35	162 (41.6)	227 (58.4)	389 (30.5)
Missing	24	44	68
**COPD**			
Yes	54 (29.2)	131 (70.8)	185 (14.5)
No	316 (33.5)	628 (66.5)	944 (73.9)
Missing	53	95	148
**Heart Failure**			
Yes	30 (18.6)	131 (81.4)	161 (12.6)
No	338 (35.0)	627 (65.0)	965 (75.6)
Missing	55	96	151

**Table 2 geriatrics-05-00058-t002:** Prevalence of Clinical Frailty Scale by Age Group and Sex in 1277 older patients admitted to 13 hospitals in COPE study.

Clinical Frailty Scale												
Number * (%)	1	2	3	4	5	6	7	8	9	Missing	Total	Increasing Frailty across Age Group (Test for Trend)
**Age**												
65–74	12 (3.3)	37 (10.1)	95 (25.9)	56 (15.3)	35 (9.6)	43 (11.7)	53 (14.5)	20 (5.5)	10 (2.7)	5	366	Reference
75–84	3 (0.6)	16 (3.1)	57 (11.2)	77 (15.1)	92 (18.0)	112 (22.0)	108 (21.2)	29 (5.7)	11 (2.2)	5	510	<0.00001
85+	0	10 (2.5)	17 (4.2)	24 (6.0)	59 (14.7)	117 (29.2)	128 (31.9)	31 (7.7)	6 (1.5)	9	401	<0.00001
**Sex ^&^**												**Increasing Frailty Male vs. Female**
Female	4 (0.7)	25 (4.2)	51 (8.5)	55 (9.2)	83 (13.8)	138 (23.0)	177 (29.5)	40 (6.7)	14 (2.3)	14	601	
Male	11 (1.6)	38 (5.6)	118 (17.4)	102 (15.1)	102 (15.1)	134 (19.9)	112 (16.6)	40 (5.9)	13 (1.9)	5	675	<0.001
Total	15	63	169	157	186	272	289	80	27	19	1276	

* 19 patients had missing CFS values; ^&^ one patient had missing sex value.

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
