# Peer review of "The Clinical Frailty Scale: Estimating the Prevalence of Frailty in Older Patients Hospitalised with COVID-19. The COPE Study"

_geriatrics, 2020, doi:10.3390/geriatrics5030058_

Round 1

Reviewer 1 Report

The manuscript is well written, but methodology and description of results should be improved. Here below you find the revisions that I suggest to improve the readibility and the quality of the paper:

-Introduction: at line 57, you could describe other reasons that support the importance of frailty assessment in addition to prevention of negative outcomes (e.g. allow a better allocation of care resources to frailer patients at higher risk of negative outcomes, avoid excessive hospital admissions which turned out to be amplifiers of the COVID-19 pandemic worldwide).

-Methods and Results : these sections need to be improved. In the Participants section you should eventually report the number of excluded patients because of missing data (if any).  ùIn the Variables paragraph, I suggest to describe that you are using categorical variables expressed in terms of number and percentage (%) as shown in table 1. At Line 85, please specify which equation you used to calculate eGFR. Additionally, in the Data Analysis section, I suggest to assess the presence of a statistically significant between-group difference for any categorical variable (demographics, comorbidities) that you considered, reporting p-value in table 1 and describing significant and non-significant results in the appropriate section.   At line 119, report overall prevalence of CKD among major comorbidities in frail and non-frail patients. 

Author Response

We thank the Reviewer for taking the time to read and publish their very helpful comments. We will address these in a point-by-point basis using the comments below (Our rebuttal in bold italics):

-Introduction: at line 57, you could describe other reasons that support the importance of frailty assessment in addition to prevention of negative outcomes (e.g. allow a better allocation of care resources to frailer patients at higher risk of negative outcomes, avoid excessive hospital admissions which turned out to be amplifiers of the COVID-19 pandemic worldwide).

  • We have included these descriptions to line 66 (was 57), as suggested (please see the revised document, in Track Changes) 

-Methods and Results : these sections need to be improved. In the Participants section you should eventually report the number of excluded patients because of missing data (if any). 

  • There were no missing data 

ùIn the Variables paragraph, I suggest to describe that you are using categorical variables expressed in terms of number and percentage (%) as shown in table 1.

  • Done 

At Line 85, please specify which equation you used to calculate eGFR.

  • MDRD, also annotated in the text

Additionally, in the Data Analysis section, I suggest to assess the presence of a statistically significant between-group difference for any categorical variable (demographics, comorbidities) that you considered, reporting p-value in table 1 and describing significant and non-significant results in the appropriate section.   At line 119, report overall prevalence of CKD among major comorbidities in frail and non-frail patients. 

  • We appreciate the reviewer's comment about significant between-group differences for the categorical variables. While we did consider performing this analysis, the objective of this short report was to describe in detail the prevalence of frailty categories in hospitalised patients with COVID-19, and the prevalence of major co-morbidities. While we captured data on admission eGFR, we did not look specifically at CKD so unfortunately we cannot comment on this. 

Again we thank the Reviewer for their very helpful comments which no doubt have improved the quality of our manuscript enormously. 

Reviewer 2 Report

Manuscript Geriatrics-900718

Abstract

Overall, well-written and clear.

A few minor points to consider:

I would temper the term ‘key criterion’. Frailty and the CFS in particular are part of a multi-dimensional assessment of older people, which can help support critical care decisions, see Hubbard RE, Maier AB, Hilmer SN, Naganathan V, Etherton-Beer C, Rockwood K. Frailty in the Face of COVID-19. Age and ageing. 2020 May 6 and O’Caoimh R, Kennelly S, Ahern E, O’Keeffe S, Ortuño RR. COVID-19 and the Challenges of Frailty Screening in Older Adults. The Journal of Frailty & Aging. 2020 Jun 11:1-2.

Rephrase or clarify the term ‘uptake’ – I presume this refers to the completion rate.

There is no need to include all of the frequency data (i.e. the sentence: The number (%) of patients with CFS 1-9 were 15(1.2%), 63(5.0%), 169(13.5%), 34 157(12.6%), 186(14.9%), 272(21.8%), 289(23.1%), 80(6.4%) and 46(3.6%), respectively) in the results subsection of the abstract – it is difficult to read. The prevalence by biological sex would be more informative.  

I would rephrase the conclusion presented here. It is not strictly accurate. This paper was presented as a prevalence study and it is not clear that it shows that “frail older people are likely to have COVID-19 disease severe enough to warrant hospitalization, compared to their robust counterparts”. While I appreciate that most of this older sample were frail, the prevalence of frailty among hospitalised older people approaches 50% or more in many studies (e.g. ref 25 in your paper indicates that 45% of patients in studies in non-community-based samples were frail). The decision-making that led to or warranted hospitalisation is not under evaluation here and it is likely that many of the frailest adults (those with CFS scores of 8 or 9) were not admitted to hospital, particularly those in nursing homes. This has been the experience in many countries including those in Europe and particularly in Italy. The authors can only claim that there was a high prevalence of frailty in this cohort. The authors do mention that selection bias likely influenced there results, which is clearly the case as it is with similar studies examining frailty and COVID in acute care settings.

Main text:

Again, well-written and clear.

Introduction:

The objectives mentioned in the abstract should also be mentioned here at the end of the introduction. I only note one (estimating the prevalence of frailty).

Methods:

Typo on line 78: using ‘the’ CFS

Line 89: The CFS also takes cognitive status into account – please correct.

Define completion rate – by whom? Was it mandatory to score the CFS at these sites during this study - if not the completion rates are staggering - hard to believe! I have conducted frailty studies where it was near impossible to get staff to complete additional tests, particularly in ED/ED triage. 

Some information on how, where and by whom the CFS was scored would be important to understand whether it is accurate – a potential limitation of scoring what is quite a subjective scale. Was inter-rater reliability (IRR) testing performed? These can all affect prevalence rates. Please present IRR results or comment on same.

Results:

Table 1 – I suggest adding a column calculating whether there were statistically significant differences between groups. This will add to the richness of the data, which are currently a little bare.

Discussion and conclusion:

Again as with my comment in the abstract, please temper down the statement that “ These data demonstrate that it is the frailer proportion of older people who need to take extra precautions for developing COVID-19, as the majority of non-frail older people will likely not have severe enough disease to warrant hospitalization.” I do not see that the results can infer this.

Typo on line 134 – ‘the’ CFS is an…

Many disagree that the CFS does not require training (see my references in the abstract) that. The test is subjective and diagnostic accuracy rates for mortality using the CFS rarely exceed 70% - which is considered at best ‘fair’ diagnostic accuracy.

As frailty proportions with COVID-19 are similar to prevalence amongst acute presentations of other acute illness amongst older adults, what additional information does the CFS provide other than acutely/gravely ill older people have higher mortality and hence are less likely to survive COVID-19 should they be so unwell as to require critical care. The argument laid out between lines 150-153 doesn’t sit well with me. Surely the point is that the higher mortality reflects the severity of the condition, hence there was lower mortality in a frail non-COVID sample. This whole paragraph is speculation and is trying to imply cause and effect – frailty causes COVID. I appreciate the authors use the word ‘if’ but strongly advise re-wording this paragraph or removing it. I am not sure how it aligns with the objectives of the paper.

Please also rephrase the conclusion – that frailty was easy to perform. This has not been assessed. If frailty screening is usually or currently mandated during the COVID-19 pandemic, than scoring should be high. This does not mean that it is easy to score. Again I direct you to Hubbard RE, Maier AB, Hilmer SN, Naganathan V, Etherton-Beer C, Rockwood K. Frailty in the Face of COVID-19. Age and ageing. 2020 May 6 and O’Caoimh R, Kennelly S, Ahern E, O’Keeffe S, Ortuño RR. COVID-19 and the Challenges of Frailty Screening in Older Adults. The Journal of Frailty & Aging. 2020 Jun 11:1-2. The poor inter-rater reliability of the CFS, particularly on hospital admission reflects how difficult it is to score on the go without a detailed collateral and without a CGA having been completed and especially when patients are unwell – the CFS is not supposed to be an eyeball test, even if it is being used in this fashion. There are a number of papers showing this and I am aware of a recent studies showing how poor its diagnostic accuracy and reliability is when it is used in ED triage (again understanding where and how is was scored will help clarify this somewhat).  

Author Response

We thank this Reviewer for taking much time and effort to read our paper and provide us with some very helpful comments, which we will address in a point-by-point fashion here in bold and italics, for clarity: 

I would temper the term ‘key criterion’. Frailty and the CFS in particular are part of a multi-dimensional assessment of older people, which can help support critical care decisions, see Hubbard RE, Maier AB, Hilmer SN, Naganathan V, Etherton-Beer C, Rockwood K. Frailty in the Face of COVID-19. Age and ageing. 2020 May 6 and O’Caoimh R, Kennelly S, Ahern E, O’Keeffe S, Ortuño RR. COVID-19 and the Challenges of Frailty Screening in Older Adults. The Journal of Frailty & Aging. 2020 Jun 11:1-2.

Term replaced, as suggested and papers cited - see discussion. 

Rephrase or clarify the term ‘uptake’ – I presume this refers to the completion rate.

Done.

There is no need to include all of the frequency data (i.e. the sentence: The number (%) of patients with CFS 1-9 were 15(1.2%), 63(5.0%), 169(13.5%), 34 157(12.6%), 186(14.9%), 272(21.8%), 289(23.1%), 80(6.4%) and 46(3.6%), respectively) in the results subsection of the abstract – it is difficult to read. The prevalence by biological sex would be more informative.  

Thank you for this helpful point, this is now amended. 

I would rephrase the conclusion presented here. It is not strictly accurate. This paper was presented as a prevalence study and it is not clear that it shows that “frail older people are likely to have COVID-19 disease severe enough to warrant hospitalization, compared to their robust counterparts”. While I appreciate that most of this older sample were frail, the prevalence of frailty among hospitalised older people approaches 50% or more in many studies (e.g. ref 25 in your paper indicates that 45% of patients in studies in non-community-based samples were frail). The decision-making that led to or warranted hospitalisation is not under evaluation here and it is likely that many of the frailest adults (those with CFS scores of 8 or 9) were not admitted to hospital, particularly those in nursing homes. This has been the experience in many countries including those in Europe and particularly in Italy. The authors can only claim that there was a high prevalence of frailty in this cohort. The authors do mention that selection bias likely influenced there results, which is clearly the case as it is with similar studies examining frailty and COVID in acute care settings.

Thank you for this insight, we have re-worded this section to clarify our meaning.

Again, well-written and clear.

We are very appreciative of this compliment.  

Introduction:

The objectives mentioned in the abstract should also be mentioned here at the end of the introduction. I only note one (estimating the prevalence of frailty).

This is an important observation, thank you for mentioning it. We have amended the text as suggested. 

Methods:

Typo on line 78: using ‘the’ CFS

Line 89: The CFS also takes cognitive status into account – please correct.

Both done (now line 105)

Define completion rate – by whom? Was it mandatory to score the CFS at these sites during this study - if not the completion rates are staggering - hard to believe! I have conducted frailty studies where it was near impossible to get staff to complete additional tests, particularly in ED/ED triage. 

Some information on how, where and by whom the CFS was scored would be important to understand whether it is accurate – a potential limitation of scoring what is quite a subjective scale. Was inter-rater reliability (IRR) testing performed? These can all affect prevalence rates. Please present IRR results or comment on same.

Thank you for sharing your experience of CFS completion rates. We have amended the text for clarity on this point. The principal investigator at each site trained each clinical investigator capturing data to use the CFS. The fact that the NICE Rapid Guidelines for COVID-19 was that CFS be assessed in all patients presenting to hospital, meant that clinicians became much more used to assessing this quickly, and the PI oversaw CFS assessment for quality control. We did not perform IRR testing, and we have added this important observation in the last paragraph in the Discussion. 

Results:

Table 1 – I suggest adding a column calculating whether there were statistically significant differences between groups. This will add to the richness of the data, which are currently a little bare.

We opted not to do this as we wanted this paper to be mainly descriptive, and that the emphasis should be on frailty and age and sex distributions in hospitalised patients with COVID-19. We will certainly consider this point for future analyses and papers. 

Discussion and conclusion:

Again as with my comment in the abstract, please temper down the statement that “ These data demonstrate that it is the frailer proportion of older people who need to take extra precautions for developing COVID-19, as the majority of non-frail older people will likely not have severe enough disease to warrant hospitalization.” I do not see that the results can infer this.

Noted, and omitted from the text. 

Typo on line 134 – ‘the’ CFS is an…

Done 

Many disagree that the CFS does not require training (see my references in the abstract) that. The test is subjective and diagnostic accuracy rates for mortality using the CFS rarely exceed 70% - which is considered at best ‘fair’ diagnostic accuracy. 

We agree with the Reviewer that the CFS can be subjective. However, to clarify our meaning, there is no specific test or methods that need to be applied to CFS scoring, apart from information about function and cognition. We have clarified this in the text. 

As frailty proportions with COVID-19 are similar to prevalence amongst acute presentations of other acute illness amongst older adults, what additional information does the CFS provide other than acutely/gravely ill older people have higher mortality and hence are less likely to survive COVID-19 should they be so unwell as to require critical care. The argument laid out between lines 150-153 doesn’t sit well with me. Surely the point is that the higher mortality reflects the severity of the condition, hence there was lower mortality in a frail non-COVID sample. This whole paragraph is speculation and is trying to imply cause and effect – frailty causes COVID. I appreciate the authors use the word ‘if’ but strongly advise re-wording this paragraph or removing it. I am not sure how it aligns with the objectives of the paper.

We take on board this observation, and have done so and amended the text for clarity. 

Please also rephrase the conclusion – that frailty was easy to perform. This has not been assessed. If frailty screening is usually or currently mandated during the COVID-19 pandemic, than scoring should be high. This does not mean that it is easy to score. Again I direct you to Hubbard RE, Maier AB, Hilmer SN, Naganathan V, Etherton-Beer C, Rockwood K. Frailty in the Face of COVID-19. Age and ageing. 2020 May 6 and O’Caoimh R, Kennelly S, Ahern E, O’Keeffe S, Ortuño RR. COVID-19 and the Challenges of Frailty Screening in Older Adults. The Journal of Frailty & Aging. 2020 Jun 11:1-2.

We have cited paper by Hubbard et al, and O'Caoimh et al, and we have rephrased the conclusion as suggested. 

The poor inter-rater reliability of the CFS, particularly on hospital admission reflects how difficult it is to score on the go without a detailed collateral and without a CGA having been completed and especially when patients are unwell – the CFS is not supposed to be an eyeball test, even if it is being used in this fashion. There are a number of papers showing this and I am aware of a recent studies showing how poor its diagnostic accuracy and reliability is when it is used in ED triage (again understanding where and how is was scored will help clarify this somewhat).  

We have tried to be more descriptive of the scoring of CFS in the Methods section, bearing in mind the word limit of this Brief Report. All our patients were admitted to hospital, therefore would have had a more comprehensive assessment than would occur in an ED triage setting. 

We thank the reviewer again for their very valuable comments, and we are confident that these changes have improved the paper enormously. 

Reviewer 3 Report

As the medical director of a long term care facility that had 246 covid cases I did use the CFS to evaluate each resident for a care plan.  The mild to moderate residents were evaluated by an expert team to determine if they would benefit from hospital.  The remaining 75% of our 485 population had severe frail or higher so they were treated onsite.

Here is my concern about the conclusion.

  1. To see frail seniors end up in the ER and hospital is more a reflection on lack of community care to support frail seniors.  In our facility we only sent in the mild or moderate frail which were very few.  Also, we had many deaths with less frail individuals.  But the point being is that frailty is made worse by the system not just the infection.
  2. Also, once infected and ill, and then measure the CFS you are assessing their frailty in a sick state not there baseline.  So it isnt clear that the frailty level assessed in hospital was not their true frail state but rather their frailty during a very sick state.

Otherwise the study is very interesting and so I would get the authors to broaden their assessment of other reasons for this outcome.

Author Response

We thank the Reviewer for taking the time to read our paper, and to give helpful comments. We will address these comments on a point-by-point basis (in bold and italics for clarity):

Here is my concern about the conclusion.

  1. To see frail seniors end up in the ER and hospital is more a reflection on lack of community care to support frail seniors.  In our facility we only sent in the mild or moderate frail which were very few.  Also, we had many deaths with less frail individuals.  But the point being is that frailty is made worse by the system not just the infection.

We take this point on board, and have amended the discussion to include this.

  1. Also, once infected and ill, and then measure the CFS you are assessing their frailty in a sick state not there baseline.  So it isnt clear that the frailty level assessed in hospital was not their true frail state but rather their frailty during a very sick state.

All patients included in this study had a CFS performed on admission, which assessed their frailty scale score 2 weeks prior to admission. This is stated in line 106 under Methods. 

Otherwise the study is very interesting and so I would get the authors to broaden their assessment of other reasons for this outcome.

Again, we thank the Reviewer for their regard for our paper and for taking the time to provide helpful comments which have no doubt improved the quality of our manuscript. 

Round 2

Reviewer 2 Report

Thank you for addressing my comments/suggestions. The paper reads well.